# Decreased Expression of Aquaporins as a Feature of Tubular Damage in Lupus Nephritis

**DOI:** 10.3390/cells14050380

**Published:** 2025-03-05

**Authors:** Maxime Melchior, Marie Van Eycken, Charles Nicaise, Thomas Duquesne, Léa Longueville, Amandine Collin, Christine Decaestecker, Isabelle Salmon, Christine Delporte, Muhammad Soyfoo

**Affiliations:** 1Departement of Rheumatology, Erasme-HUB Hospital, Université Libre de Bruxelles, 1070 Brussels, Belgium; maxime.melchior@ulb.be; 2Department of Pathology, Erasme-HUB Hospital, Université Libre de Bruxelles, 1050 Brussels, Belgium; marie.van.eycken@ulb.be (M.V.E.); isabelle.salmon@hubruxelles.be (I.S.); 3URPhyM, NARILIS, Université de Namur, 5000 Namur, Belgium; charles.nicaise@unamur.be (C.N.); thomas.duquesne@unamur.be (T.D.); lea.longueville@unamur.be (L.L.); 4DIAPath, Center for Microscopy and Molecular Imaging, Université Libre de Bruxelles, 6041 Gosselies, Belgium; amandine.collin@ulb.be (A.C.); christine.decaestecker@ulb.be (C.D.); 5Laboratory of Image Synthesis and Analysis, Université Libre de Bruxelles, 1050 Brussels, Belgium; 6Centre Universitaire Inter Régional D’expertise en Anatomie Pathologique Hospitalière, 6040 Jumet, Belgium; 7Laboratory of Pathophysiological and Nutritional Biochemistry, Faculty of Medicine, Université Libre de Bruxelles, 1070 Brussels, Belgium; christine.delporte@ulb.be

**Keywords:** lupus nephritis, lupus, aquaporin 1, aquaporin 2, aquaporin 3, aquaporins

## Abstract

**Background:** Tubulointerstitial hypoxia is a key factor for lupus nephritis progression to end-stage renal disease. Numerous aquaporins (AQPs) are expressed by renal tubules and are essential for their proper functioning. The aim of this study is to characterize the tubular expression of AQP1, AQP2 and AQP3, which could provide a better understanding of tubulointerstitial stress during lupus nephritis. **Methods:** This retrospective monocentric study was conducted at Erasme-HUB Hospital. We included 37 lupus nephritis samples and 9 healthy samples collected between 2000 and 2020, obtained from the pathology department. Immunohistochemistry was performed to target AQP1, AQP2 and AQP3 and followed by digital analysis. **Results:** No difference in AQP1, AQP2 and AQP3 staining location was found between healthy and lupus nephritis samples. However, we observed significant differences between these two groups, with a decrease in AQP1 expression in the renal cortex and in AQP3 expression in the cortex and medulla. In the subgroup of proliferative glomerulonephritis (class III/IV), this decrease in AQPs expression was more pronounced, particularly for AQP3. In addition, within this subgroup, we detected lower AQP2 expression in patients with higher interstitial inflammation score and lower AQP3 expression when higher interstitial fibrosis and tubular atrophy were present. **Conclusions:** We identified significant differences in the expression of aquaporins 1, 2, and 3 in patients with lupus nephritis. These findings strongly suggest that decreased AQP expression could serve as an indicator of tubular injury. Further research is warranted to evaluate AQP1, AQP2, and AQP3 as prognostic markers in both urinary and histological assessments of lupus nephritis.

## 1. Introduction

Systemic lupus erythematosus (SLE) is a heterogeneous disease, with significant variability in clinical presentation and organ involvement. Among its complications, lupus nephritis (LN) represents a distinct and severe subgroup, affecting up to one-third of SLE patients, with 5 to 20% progressing to end-stage kidney disease (ESKD) [1]. Compared to non-LN SLE patients, those with LN face a higher burden of morbidity, requiring more aggressive immunosuppressive therapy and closer long-term monitoring [2,3]. Despite advances in our understanding of SLE, LN remains a major therapeutic challenge, with persistent gaps in predicting disease progression and response to treatment. These differences underscore the need for personalized treatment strategies tailored to the unique pathophysiology and prognostic factors of LN, optimizing both renal and overall patient outcomes [3,4].

Many factors responsible for glomerular and tubular inflammation of LN have been identified including immune complexes, local complement activation, leukocytes recruitment and cytokine signaling pathways [3]. However, little data are available on the mechanisms regulating inflammation within renal parenchyma. Indeed, persistent and on-going inflammation, particularly with the occurrence of tubulointerstitial immune infiltrates appears to be a key element in the progression from LN to ESKD [3].

Aquaporins (AQPs) are transmembrane proteins that are widely distributed among cells in the human body [5]. Thirteen human AQPs were described and classified into three sub-categories: classical aquaporins (AQP0, AQP1, AQP2, AQP4, AQP5, and AQP8) promoting the transport of water and small ionized molecules; aquaglyceroporins (AQP3, AQP7, AQP9, and AQP10) permeable to water and other soluble molecules with a higher molecular weight (urea, glycerol); atypical aquaporins (AQPs 11 and 12) for which the detailed function remains to be defined [6,7] although AQP11 was shown to be permeable to glycerol as well [8]. Eight AQPs are expressed in the kidney (AQPs 1–7 and 11) and several of which are involved in inflammatory and degenerative processes, including AQPs 1, 2, 3 and 11 [6,7,9].

AQP1 is the first aquaporin described and is expressed in many organs such as choroid plexus, inner ear, pancreas, liver, skin and kidney [5]. It allows the non-selective transmembrane transport of water, small monovalent cations in a cGMP-mediated manner [9], and NH_3_ and CO_2_ gases [10]. Within the nephron, it is strongly expressed by parietal cells of proximal tubule and descending Henle loop, and endothelial cells of descending vasa recta [11]. It participates to urinary concentration independently of vasopressin [12]. Through its expression on endothelial cells and macrophages, it also participates in the mechanisms of angiogenesis, cell migration and differentiation [9,13]. A protective role of AQP1 was observed in a mouse model of acute renal injury (AKI) induced by ischemia-reperfusion [14]. A recent study suggests that this effect might be related to an AQP1-dependent polarization of renal macrophages into the M2 phenotype under regulation of the PI3K/AKT/mTOR pathway [15]. A cytoprotective role of AQP1 was also shown in vitro on cells derived from the proximal tubule (HK-2) [16]. Decreased urinary excretion of AQP1-containing extracellular vesicles has been described in several mouse models of AKI, suggesting its use as a urinary biomarker of renal impairment [14,17].

AQP2 is almost exclusively expressed at the apical side of the kidney connecting tubules and collecting ductal tubule cells [18]. It is a water-selective channel involved in urine concentration in conjunction with AQP3 and AQP4 located on the basolateral side of the same cells. Vasopressin, through its V2 receptor, induces the trafficking of intracellular AQP2-containing vesicles to the apical plasma membrane (short-term regulation) and the transcription of AQP2 gene (long-term regulation) [6,9]. AQP2 is extensively studied as a urinary biomarker due to the presence of AQP2-containing extracellular vesicles in urine. A decrease in AQP2 urinary excretion has been linked to NF-kB pathway regulation in an LPS-induced AKI model [19]. In a renal transplant model, the initially observed decrease in AQP2 excretion was corrected by cyclosporine treatment [20]. In humans, the increase in urinary concentration after treatment with a vasopressin analogue could be used a marker of therapeutic response in diabetes insipidus [21].

AQP3, widely distributed among human cells, is expressed on the basolateral side of the principal cells of the collecting ducts. This aquaglyceroporin promotes the transport of glycerol, urea and hydrogen peroxide, conferring a protective role against oxidative stress and cellular metabolic input [9,13]. In a mouse model of ischemia-reperfusion, mice disabled for AQP3 showed an increased inflammatory response and greater structural damage, mainly by an increased apoptotic phenomenon [22].

To our knowledge, few studies have investigated the expression of AQPs in lupus nephritis. A 2003 study performed using several types of glomerulonephritis including 4 lupus nephritis suggested a decrease in the expression of AQP2 and AQP3 and an increase in the expression of AQP1 in pathological glomeruli [23]. We aimed to further study the expression of AQP1, AQP2 and AQP3 in the nephrons of patients with LN and its possible association with histological/clinical activity or chronicity.

## 2. Material and Methods

### 2.1. Patient Selection and Clinical Data Collection

This single-center retrospective study was conducted at Erasme-HUB Hospital in Brussels and was approved by the ULB-Erasme ethics committee (approval number: P2020/710) in December 2020. We started by screening the samples available in the pathology department’s biobank. We identified a total of 48 renal needle biopsy samples taken from patients treated in our hospital between 2000 and 2020. Of these, 11 were excluded due to insufficient remaining material on the archived paraffin-embedded block of needle kidney biopsy. As a result, 37 LN samples were used in this study.

Additionally, we selected 9 healthy biopsy samples archived in the pathology department’s biobank. These specimens were collected as part of a standard organ transplantation procedure (5 out of the 9 samples) or in a macroscopically healthy renal area (4 out of the 9 samples) following nephrectomy for renal cancer.

Clinical data were obtained from patient’s medical records with a focus on the following characteristics: age, sex, calculation of the systemic lupus erythematosus disease activity index 2000 (SLEDAI-2K), proteinuria at diagnosis, proteinuria at 3 months, proteinuria at 6 months, proteinuria at 1 year, estimated glomerular filtration rate (eGFR) at diagnosis, eGFR at 1 year, white blood cell count (WBC), hemoglobinemia, platelet count, neutrophil count, lymphocyte count, erythrocyte sedimentation rate (ESR), C reactive protein (CRP), urea, creatinine, complement component (C3, C4), antinuclear antibodies titer (ANA titer), antibodies anti-double-stranded DNA (anti-dsDNA), anti-nucleosomes, anti-Smith (anti-Sm), the presence or absence of any antiphospholipid antibody or lupus anticoagulant, leukocyturia, hematuria and albuminuria at diagnosis. All laboratory tests on blood and urine samples were carried out in accordance with LHUB-ULB’s standard clinical practice procedures.

Renal biopsy sections were reviewed by a pathologist with expertise in renal diseases, and lupus nephritis samples were categorized by class, scored with activity index (0–24; composite index with endocapillary hypercellularity, neutrophils/karyorrhexis, fibrinoid necrosis, hyaline deposits, cellular/fibrocellular crescents and interstitial inflammation) and chronicity index (0–12; composite index with total glomerulosclerosis score, fibrous crescents, tubular atrophy and interstitial fibrosis) according to the ISN/RPS 2018 guidelines [24].

### 2.2. Immunohistochemistry Staining

Automated immunohistochemistry (IHC) on 4 µm-thick formalin-fixed paraffin-embedded (FFPE) kidney biopsy sections were processed on a PT Link and an Autostainer Link 48 (Agilent Technologies, Santa Clara, CA, USA). Prior to staining, FFPE tissue sections were subjected to deparaffinization, and hydration followed by heat-induced epitope retrieval, except for AQP1, in Dako Target Retrieval Solution pH9 (K800421-2, Agilent Technologies Inc.) 30 min at 97 °C. The sections were incubated with AQP2 (HPA046834, dilution 1/1000, Sigma-Aldrich, St Louis, MI, USA) and AQP3 (HPA0149924, dilution 1/1000, Sigma-Aldrich) antibodies for 30 min at room temperature. The incubation of AQP1 (HPA019206, dilution 1/1700, Sigma-Aldrich) for 40 min at room temperature was followed by the incubation of Envision Flex+ Rabbit (K800921-2; Agilent Technologies Inc.). All the sections were then incubated with Dako Envision Flex detection system (GV800; Agilent Technologies, Santa Clara, CA, USA) according to the manufacturer’s protocol. The sections were counterstained with hematoxylin (K800821-2; Agilent Technologies). Kidney control tissue and universal negative control antibody (omissions of the primary antibody) were processed in parallel, using the protocol as described above.

### 2.3. Digital Analysis

Slide images were acquired at 20× magnification using the C9600-12 NanoZoomer digital slide scanner (Hamamatsu, Shizuoka, Japan) with a resolution of 453 nm per pixel. Subsequent image analysis was performed using the open-source imaging software NIH-ImageJ (Fiji, version 2.3) by two independent investigators (TD and LL), single-blinded to experimental groups. Regions of interest were defined by manual contouring based on morphological differences between the cortex and medulla. A sub-analysis of the cortex focusing on glomeruli was carried out for AQP1 labeling.

For the assessment of AQP expression, we employed color deconvolution with the ‘H&DAB’ vector, effectively separating 3,3′-diaminobenzidine (DAB) and hematoxylin (H) staining. In the color channel 2 (representing DAB staining for aquaporin), we optimized the threshold within the range of 0 to 160. Multiple regions-of-interest were examined, including glomeruli (specifically for AQP1 staining), encompassing a minimum of five entire, high-quality glomeruli, and cortical or medullary regions spanning a representative area exceeding 5000 pixels. The data obtained is reported as a percentage of labeling within the defined region of interest (%ROI).

### 2.4. Statistical Analysis

All data were tested for normality using the Shapiro–Wilk test. Based on the results, non-parametric tests were applied. A comparison of sex distribution between cases and controls was tested by Fisher’s exact test. For single comparisons, such as %ROI differences between LN and controls, NIH score subsets, and eGFR at one year, none of the datasets followed a normal distribution; therefore, Mann–Whitney tests were used. For multiple comparisons, including %ROI differences among LN subsets, none of the datasets followed a normal distribution; therefore, group differences were assessed using the Kruskal–Wallis test followed by post hoc Dunn’s tests. Data are presented as medians with interquartile ranges (Quartile 1–Quartile 3). Categorical variables are reported as counts with percentages. Correlation analysis was conducted using Spearman’s correlation test, as AQP3 %ROI data did not follow a normal distribution. Statistical analyses were performed using Prism 8, with a significance threshold set at *p* < 0.05.

## 3. Results

### 3.1. Clinical Features

Our study encompassed a cohort of 37 patients, all of whom received confirmed diagnoses of LN with 78.4% diagnoses with LN during a flare of well-known SLE and 21.6% as prior manifestation. All except one of the patients were women (36 out of 37) and exhibited high lupus activity scores (SLEDAI-2k) with a median score of 16, as well as pathological urine analysis suggestive of renal involvement prior to undergoing a biopsy (Table 1). Within this cohort, 21.6% (8 out of 37) of the patients exhibited nephrotic range proteinuria. The majority of these individuals responded well to treatment, approximately two-thirds of them (19 out of 27) achieving 25% reduction in proteinuria within three months as recommended by EULAR [2] (Table 1). At 6 months follow-up, 42.3% (11 out of 26) were responders with a 50% reduction in proteinuria and 58% (15 out of 26) of the patients achieved a proteinuria level lower than 0.5 g/g after one-year follow-up (Table 1). Notably, two of the patients not achieving this threshold initially presented with nephrotic range proteinuria (Table 1). Thirteen percent (4 out of 30) of the patients faced end-stage renal insufficiency after one year (Table 1).

A few patients in the cohort presented comorbidities that might have impacted their kidney function: 4 out of 37 had a prior history of LN exceeding two years, 1 out of 37 had diabetes, 5 out of 37 had high blood pressure, and 2 out of 37 were active tobacco smokers (Table 1). Further clinical, demographic, and histological characteristics that might influence disease severity are detailed in Table 1.

We also selected nine control tissue biopsies from the pathology department biobank. These biopsies were taken as part of a standardized pre-transplant procedure or in a healthy area during nephrectomy for localized renal cell carcinoma (Table 2). As donor data were not included in the medical records, we could only compare age and sex with lupus samples. There was no difference in sex distribution, but the control samples were significantly older than LN samples (Appendix A, Table 2).

### 3.2. Analysis of AQP Expression

We first analyzed the distribution of each aquaporin (AQP) in healthy kidney tissue. As expected, AQP1 showed weak labeling in glomeruli but was strongly expressed on the apical surface of proximal convoluted tubules in the cortex, with weaker expression on their basolateral side (Figure 1 and Appendix A). In the medulla, AQP1 was predominantly localized to the descending loop of Henle and the vasa recta (Figure 1 and Appendix A). AQP2 was exclusively expressed on the apical membrane of collecting tubules in both the cortex and medulla (Figure 1 and Appendix A). AQP3 was absent in glomeruli, displayed weak basolateral expression in proximal convoluted tubules, and showed strong basolateral expression in distal convoluted and collecting tubules (Figure 1 and Appendix A). These findings align with previous reports on AQP localization in the kidney [7].

In lupus nephritis (LN), although we observed a reduction in staining for AQP1, AQP2, and AQP3; their localization remained unchanged. Notably, in healthy controls, no staining for AQP1, AQP2, or AQP3 was detected in the interstitium.

In contrast, LN patients with pronounced inflammatory infiltrates exhibited moderate AQP3 membrane labeling in mononuclear cells with morphological features suggestive of plasma cells (Appendix A), which are frequently observed in LN [24]. This finding is consistent with reports indicating that some plasmablasts express AQP3 [25].

To better assess differences in AQP labeling between LN samples and healthy controls, we then performed digital analysis, comparing the fraction of labeled areas in regions of interest (%ROIs), i.e., glomerulus, cortex, and medulla. Glomerular analysis was conducted only for AQP1, as the other studied aquaporins were not expressed in this region. Our results revealed a significant decrease in AQP1 labeling in the cortex but not in the medulla (Figure 2). Further analysis focusing on glomeruli showed a small but non-significant decrease in decrease in expression of AQP1, suggesting that the observed cortical reduction is meanly driven by tubular involvement (Figure 2). No differences were detected for AQP2 at either the cortical or medullary level. In contrast, AQP3 expression was significantly reduced in LN patients in both the cortical and medullary regions (Figure 2).

### 3.3. Association Between Aquaporin Expression and Tubular Damage

We next examined differences in AQP expression across LN subsets based on the 2018 ISN/RPS classification [22]. We categorized LN cases into non-proliferative subsets (classes I, II, and V), which are associated with a lower risk of renal failure, and proliferative subsets (classes III and IV), which carry a higher risk of renal failure [2,4]. Interestingly, our analysis revealed a significant decrease in AQP3 expression specifically in the proliferative LN (P-LN) subset, whereas no difference was observed in the non-proliferative LN (NP-LN) subset (Figure 3). In contrast, AQP1 expression was reduced in both LN subsets compared to controls, but no significant difference was found between proliferative and non-proliferative groups (Figure 3).

To further explore potential associations, we analyzed the relationship between AQP expression and NIH activity and chronicity indices. No correlation was found between AQP1, AQP2, or AQP3 expression and the overall NIH activity index. However, among individual components of the activity index, a decrease in cortical AQP2 expression was specifically associated with interstitial inflammation, as indicated by leukocyte infiltration in the cortical interstitium (Figure 3). Among the NIH chronicity index components, the presence of atrophic cortical tubules and interstitial fibrosis in the superficial cortex were significantly associated with reduced AQP3 expression (Figure 3). A similar trend was observed for cortical AQP2 expression, although it did not reach statistical significance (Appendix A). Furthermore, using Spearman correlation analysis, we identified a strong negative correlation between AQP3 expression and the chronicity index (r = −0.6218; *p* < 0.01) (Appendix A).

Given the prognostic importance of tubular involvement in renal function, we further investigated potential links between AQP expression and clinical data at both diagnosis and follow-up. While no statistically significant associations were found, a higher proportion of patients with initially lower AQP2 or AQP3 expression experienced renal insufficiency at one year (Appendix A).

Altogether, our results underscore the association between decreased aquaporin-3 expression and severe renal damage linked to poor prognosis.

## 4. Discussion

Among autoimmune diseases, numerous alterations in AQP expression have been documented, notably in the salivary glands of Sjögren’s syndrome or the association with neuromyelitis optica mediated by AQP4 [13,25,26]. Eight types of AQPs are expressed in the kidney (AQPs 1–7 and 11) [18]. To our knowledge, only one other study has investigated AQP expression in LN [21]. This study examined a range of acute, chronic, immune, and non-immune glomerulonephritis, providing a broad overview of AQP alterations in different pathological contexts. However, it included only four LN cases without precise classification or clinical data. The authors reported an increase in AQP1 expression in lupus glomeruli and a decrease in AQP1, AQP2, and AQP3 expression in renal tubules based on immunohistochemistry (IHC). Tissue mRNA analysis confirmed reduced *AQP2* and *AQP3* expression but did not support the IHC findings for AQP1 [23]. Building on these initial observations, our study provides statistical confirmation of the decreased expression of AQP3 in the cortex and medulla, as well as AQP1 in proximal tubules. However, we did not replicate the reported changes in AQP2 expression or the increase in AQP1 at the glomerular level. These discrepancies may be attributed to the variability in glomerular AQP1 expression, and the larger sample size used in our study.

Additionally, we present novel findings regarding LN subsets, demonstrating a more pronounced reduction in cortical AQP3 expression in proliferative LN, which are associated with worse renal prognosis. While tubular injury is a well-established prognostic factor in LN, current assessment methods remain limited, as the NIH score in the ISN/RPS classification primarily focuses on glomerular lesions [24]. Our results suggest that AQP3 immunohistochemistry could serve as a valuable complementary tool to better assess LN severity.

Furthermore, we observed AQP3 expression in certain mononuclear cells within the interstitial inflammatory infiltrates, resembling plasmablasts commonly found in LN [24]. Further characterization of these cells and their involvement in LN pathophysiology is warranted.

In conclusion, our study revealed a reduction in the expression of AQP1 and AQP3 in kidney parenchyma of LN patients. This result strongly suggests that such a decrease may serve as a reliable indicator of tubulointerstitial damage. Hence, AQP1, AQP2 and AQP3 expressions should be explored as prognostic urinary and histological markers in LN.

## 5. Limitations and Perspectives

While our study provides valuable insights, its retrospective, monocentric design and limited sample size necessitate caution in extrapolating the results. To further validate our findings, larger studies including both immune and non-immune nephropathies are needed. Indeed, Bajema et al. emphasized the non-specific yet noteworthy nature of these modifications [24]. Additionally, our study included a significant age difference between LN patients and controls. However, we found no correlation between AQP expression and age, and existing literature suggests that AQP expression might decrease with aging [27]. If age had any effect, it would likely have led to an underestimation of the differences we observed. Although our study uses a more objective method of quantification than qualitative assessment, our method can be improved by further analysis at the cellular level or by including a count of the number of tubules per slide. Another limitation of our study is that, due to tissue availability, we focused on only three aquaporins. However, other AQPs warrant investigation, such as aquaporin 11, a member of the superaquaporin subfamily, which has been genetically linked to renal diseases in both human studies and mouse models [6,25]. Additionally, aquaporins 4 and 5, which are expressed in the kidney, have been implicated in systemic connective tissue diseases affecting other organs [25]. Furthermore, given previous findings in IgA nephropathy, analyzing urinary AQP excretion during the follow-up of lupus patients, both with and without LN, could provide valuable insights into disease progression and renal involvement [27].

## Figures and Tables

**Figure 1 cells-14-00380-f001:**
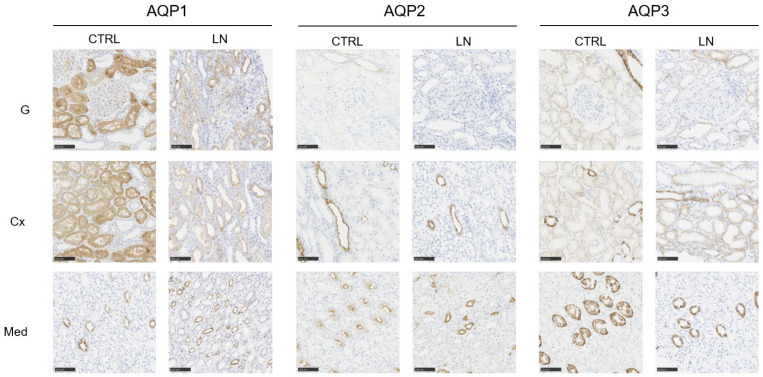
Representative staining of AQP1, AQP2 and AQP3 in lupus nephritis (LN) and healthy (CTRL) kidney tissue with a focus on glomerular (G), cortical (Cx), or medullary (Med) structures (field magnification 200×).

**Figure 2 cells-14-00380-f002:**
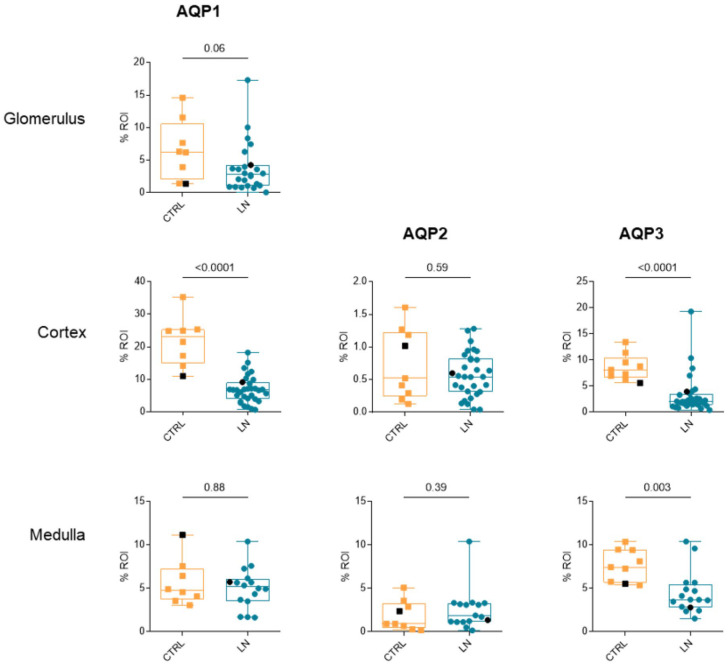
Semiquantitative analysis of AQP1, AQP2 and AQP3 staining in lupus nephritis (LN) and healthy (CTRL) kidney tissue. Results are displayed with box-and-whisker plots (min-max-median–Q1–Q3) and overlaying dots for individual data points (male patient in black). Data are expressed as percentage of labeled area within the region-of-interest (%ROI) (AQP1 Cortex: n = 8 for CTRL, n = 32 for LN; AQP1 Medulla: n = 8 for CTRL, n = 15 for LN; AQP2 cortex: n = 9 for CTRL, n = 31; AQP2 medulla: n = 9 for CTRL, n = 15; AQP3 cortex: n = 9 for CTRL, n = 33 for LN; AQP3 medulla: n = 9 for CTRL, n = 15 for LN; tested by Mann–Whitney).

**Figure 3 cells-14-00380-f003:**
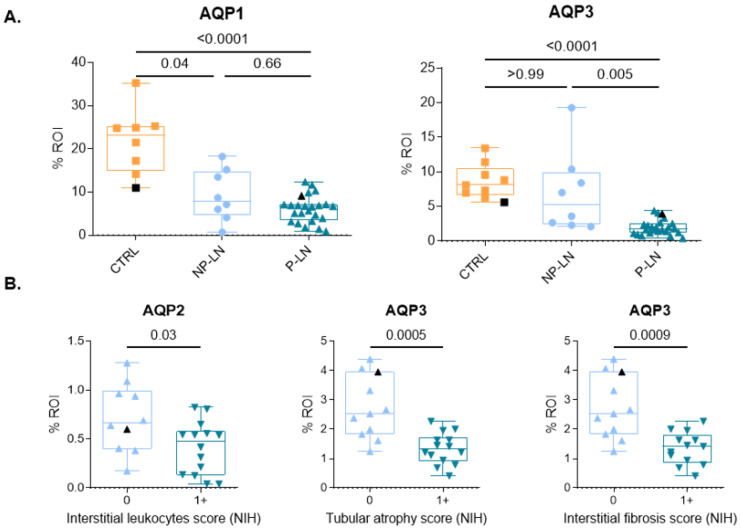
Subgroup analysis of AQPs expression in lupus nephritis. (**A**) Proportion of staining in renal cortex (%ROI) according to proliferative LN (P-LN; Class III and IV considered as proliferative) or non-proliferative LN (NP-LN; Class I, II or V considered as non-proliferative) (**B**) Proportion of staining in renal cortex of proliferative LN (%ROI) according to absence (0) or presence (score of 1 or more) of tubular or interstitial damage according to the NIH LN activity and chronicity scoring system. Only significant results are shown. Results are displayed with box-and-whisker plots (min-max-median–Q1–Q3) and overlaying dots for individual data points (male patient in black). Data are expressed as percentage of labeled area within the region-of-interest (%ROI) ((**A**) AQP1 n = 8 for ctrl, n = 8 for NP-LN, n = 24 for P-LN, tested by Kruskal–Wallis with Dunn’s multiple comparison test; AQP3 n = 8 for ctrl, n = 8 for NP-LN, n = 24 for P-LN, tested by Kruskal–Wallis with Dunn’s multiple comparison test. (**B**) AQP2 interstitial leukocytes score n = 24, n = 10 for score 0, n = 15 for score 1 or more; AQP3 interstitial leukocytes score n = 24, n = 11 for score 0, n = 13 for score 1+; AQP3 tubular atrophy score n = 24, n = 11 for score 0, n = 13 for score 1; tested by Mann–Whitney).

**Table 1 cells-14-00380-t001:** Clinical and histological data from SLE patients (expressed as median with interquartile range). SLEDAI-2k, systemic lupus erythematosus disease activity index 2000; eGFR, estimated glomerular filtration rate; WBC, white blood cell; RBC, red blood cell; ESR, erythrocytes sedimentation rate; CRP, C reactive protein; LN, lupus nephritis; NIH, National Institutes of Health; ISN/RPS, International Society of Nephrology/Renal Pathology Society.

Clinical Data at Diagnosis		
Variable (n)	Median [Q1–Q3]	Normal Range (Unit)
Age (37)	35 [28–41]	(years)
Female/Male ratio (37)	36/1	
Previous LN (>2 y)	10.8% (4/37)	
Diabetes	2.7% (1/37)	
Hypertension	13.5% (5/37)	
Tobacco	5.4% (2/37)	
Flare/New diagnosis (37)	78.4% (29/37)/21.6% (8/37)	
SLEDAI-2k (37)	16 [14–22]	
Urea (37)	35.3 [23–62]	17–48 (mg/dL)
Creatinine (37)	0.8 [0.6–1.2]	0.5–0.9 (mg/dL)
eGFR (37)	100 [50.5–112]	>90 (mL/min/1.73 m^2^)
eGFR distribution		
>90	51.4% (18/35)	
60–90	17.1% (6/35)	
15–60	25.7% (9/35)	
<15	5.7% (2/35)	
ESR (31)	53 [35–77]	<20 (mm/h)
CRP (36)	0.86 [0.43–6.90]	<5 (mg/L)
C3 (34)	58.5 [37–76.75]	80–164 (mg/dL)
C4 (35)	8 [5.5–14.5]	10–40 (mg/dL)
ANA titer (37)	1/1280 [1/640–1/2560]	<1/160
Anti-dsDNA (34)	175 [42.25–466.5]	<30 (U/mL)
Anti-Sm positivity (35)	34.3% (12/35)	
Antiphospholipids positivity (30)	26.7% (8/30)	
Leucocyturia (30)	64 [11.5–172]	<30 (WBC/µL)
Hematuria (31)	63 [24–175.5]	<30 (RBC/µL)
Urine protein-to-creatinine	2.00 [1.10–3.40]	<0.5 (g/g)
ratio (PCR) (37)		
Nephrotic range (NR)	21.6% (8/37)	
LN classification distribution (ISN/RPS 2018)		
Class		
I	2.7% (1/37)	
II	2.7% (1/37)	
III	21.6% (8/37)	
IV	40.5% (15/37)	
V	21.6% (8/37)	
III + V	5.4% (2/37)	
IV + V	5.4% (2/37)	
Class III/IV NIH indices (ISN/RPS 2018)		
NIH activity	9 [6–10]	
NIH chronicity	2 [1–4]	
Clinical data during follow-up		
Variable (n)	Median [Q1–Q3]	Normal range (units)
PCR at 3 months (27)	1 [0.65–1.70]	<0.5 (g/g)
25% decrease at 3 m	70.4% (19/27)	
PCR at 6 months	0.49 [0.15–1.07]	<0.5 (g/g)
50% decrease at 6 m	42.3% (11/26)	
PCR at 12 months	0.34 [0.10–1.04]	<0.5 (g/g)
<0.5 g/g	57.7% (15/26)	
eGFR at 12 months	100 [75–100]	>90 (mL/min/1.73 m^2^)
eGFR at 12 months distribution		
>90	60% (18/30)	
60–90	16.7% (5/30)	
15–60	6.7% (2/30)	
<15	13.3% (4/30)	

**Table 2 cells-14-00380-t002:** Clinical data from healthy controls (expressed as median with interquartile range).

Clinical Data of Controls	
Variable (n)	Median [Q1–Q3]
Age (8)	55 [43–57.5]
Female/Male ratio (8)	7/1
Comorbidities (nephrectomy samples only)	
Tobacco (4)	0% (0/4)
Diabetes (4)	0% (0/4)
Hypertension (4)	100% (4/4)
Reason for biopsy	
Prior to transplantation	55.5% (5/9)
Nephrectomy	44.4% (4/9)

## Data Availability

Data availability is restricted due to ethical restrictions.

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
