# Peer review of "Decreased Expression of Aquaporins as a Feature of Tubular Damage in Lupus Nephritis"

_cells, 2025, doi:10.3390/cells14050380_

Round 1
Reviewer 1 Report
Comments and Suggestions for Authors
1. 49 of 98 patients were excluded due to missing clinical data. The number of patients missing data was too huge. The sample size of 37 was small. The result might be affected.
2. In results section, in the sentences “two-third of them (19 out of 27) 171 achieving 25% reduction in proteinuria within three months as recommended by EULAR 172 (23) (Table S1). At 6-months follow-up, 42.3% (11 out of 26) were responder with a 50% 173 reduction in proteinuria and 58% (15 out of 26) of the patients achieved a proteinuria level 174 lower than 0.5g/g after one-year follow-up (Table S1)”, the source of the numbers, such as 27,26 was obscure.
3. The authors described the detailed distribution of AQPs. The figures with ×400 amplification is better.
4. How did the region of interest defined ?
5. The discussion would be more focused on the results of the study.
Comments on the Quality of English LanguageCan be improved.
Author Response
Dear Reviewer,
The authors thank you for your remarks and comments . Please find below the answers to your queries and comments.
R1. Comment 1: 49 of 98 patients were excluded due to missing clinical data. The number of patients missing data was too huge. The sample size of 37 was small. The result might be affected.
R1. Response 1: We thank the reviewer for highlighting this major limitation. Following this comment we have clarified patient selection in the materials and methods
“In the pathology department biobank, we identified a total of 48 renal needle biopsy samples from patients treated in our hospital between 2000 and 2020. Of these, 11 were excluded due to insufficient remaining material.”
and have further addressed this limitation in the “Limitations of the study and perspectives” section of the discussion.
“While our study provides valuable insights, its retrospective and monocentric design with a limited sample size warrants caution in extrapolating the results.”
R1. Comment 2: In results section, in the sentences “two-third of them (19 out of 27) 171 achieving 25% reduction in proteinuria within three months as recommended by EULAR 172 (23) (Table S1). At 6-months follow-up, 42.3% (11 out of 26) were responder with a 50% 173 reduction in proteinuria and 58% (15 out of 26) of the patients achieved a proteinuria level 174 lower than 0.5g/g after one-year follow-up (Table S1)”, the source of the numbers, such as 27,26 was obscure.
R1. Response 2: we agreed with the reviewer that tables 1 and S1 lacked precision. In this regards, we modified these tables to better represent the data presented and the origin of those data.
R1. Comment 3: The authors described the detailed distribution of AQPs. The figures with ×400 amplification is better.
R1. Response 3: We provided a new figure with 400x magnification in supplementary data.
R1. Comment 4: How did the region of interest defined ?
R1. Response 4: This important additional information has been added to the “Materials and methods” section.
“Regions of interest were defined by manual contouring based on morphological differences between the cortex and medulla. A sub-analysis of the cortex focusing on glomeruli was carried out for AQP1 labeling.”
R1. Comment 5: The discussion would be more focused on the results of the study.
R1. Response 5: We thank the reviewer for this comment and rewrite the discussion section accordingly.
Reviewer 2 Report
Comments and Suggestions for Authors
The manuscript submitted to the journal Cells, titled “Aquaporin 1, 2, and 3 Expressions in Lupus Nephritis: Insights into Tubular Pathology,” examines the role of aquaporins (AQP) in lupus nephritis. Although it provides clinical data on aquaporins (1, 2, and 3) in disease conditions, the study largely replicates findings previously reported by the Robert J. Walker group in 2003, specifically regarding aquaporin expression in normal human kidneys and renal diseases. Consequently, the manuscript lacks significant novelty, and the authors are encouraged to present additional data to support publication.
Author Response
According to the recommendations of the referees, we made important modifications to the text (highlighted in yellow in the revised manuscript) and the figures.
We would like to thank the reviewers and the editorial board for their constructive suggestions, and hope that you will now find our manuscript suitable for publication.
We thank you again for giving us the opportunity to improve our work.
R2. Comment 1: The manuscript submitted to the journal Cells, titled “Aquaporin 1, 2, and 3 Expressions in Lupus Nephritis: Insights into Tubular Pathology,” examines the role of aquaporins (AQP) in lupus nephritis. Although it provides clinical data on aquaporins (1, 2, and 3) in disease conditions, the study largely replicates findings previously reported by the Robert J. Walker group in 2003, specifically regarding aquaporin expression in normal human kidneys and renal diseases. Consequently, the manuscript lacks significant novelty, and the authors are encouraged to present additional data to support publication.
R2. Response 1: Our manuscript follows on from the work of Bajema et al. However, we are convinced that we have added several important elements to this first study: firstly, we have been able to test the hypotheses statistically to confirm or refute some of the observations made by this team. Secondly, we have brought new elements to bear on the prognostic significance of AQP3 staining. These aspects were not clearly emphasized in the previous version of the manuscript, and we have considerably modified the results and discussion in this respect.
Reviewer 3 Report
Comments and Suggestions for Authors
Overall, the manuscript addresses an interesting problem with significant clinical relevance. Lupus nephritis (LN) is a severe complication of systemic lupus erythematosus (SLE) associated with poor prognosis. Therefore, advancing knowledge about novel biomarkers and potential treatment options is crucial to improving outcomes for patients. Please find my detailed comments and suggestions below:
The current title suggests the paper is more of a review than an original study. I recommend making the title more specific and informative to accurately reflect the content and focus of the study.
The "Results" section is challenging to follow. Consider presenting the results step-by-step to improve readability and clarity.
Adding information in the "Conclusions" section about the clinical utility of your findings, either as biomarkers or potential therapeutic targets, could enhance the abstract's impact and clinical relevance.
In the first paragraph, emphasize the clinical differences between LN and non-LN SLE patients. Clearly articulate how these differences highlight the need for personalized treatment strategies in this subgroup.
Lines 44-46: Provide relevant references to support the data mentioned. Additionally, clarify whether "actors" or "factors" is the intended term.
Line 50: Include an appropriate reference for the statement made.
Line 60: Expand the first sentence to provide more context and include supporting references.
Use past tense consistently when describing the study's methodology (e.g., "we aimed to," "we analyzed").
Line 101: Provide the specific approval number and date for ethical approval.
Lines 105-106: Specify the source and conditions under which control kidney samples were collected.
Lines 109-118: Detail the specific methods used for measurements and analyses.
Line 157: State the type of test used to assess the normality of the data.
The "Statistics" section requires improvement. Clearly describe all statistical tests used, including their purpose and justification. For instance: Why were non-parametric tests used for correlation? Use Q1–Q3 ranges instead of "IQR."
Line 167: Replace "Most" with "All except one," if accurate.
Clarify whether the LN patients included were newly diagnosed SLE cases or those experiencing flares.
Table 1: Include the table in the main text rather than as a figure. Additionally, provide reference ranges for laboratory measurements if available.
Figure 2: Add specific p-values and ensure that statistically significant p-values are rounded to the third decimal place (or use p < 0.001 for very small values). Values > 0.05 should be rounded to the second decimal place.
Confirm whether LN patients and controls were comparable in terms of sex and age.
Improve the readability of figures and tables. Consider using box-and-whisker plots (with min, max, median, Q1, and Q3) and overlaying dots for individual data points.
Line 279: Provide more references to substantiate claims about alterations in AQP.
The discussion is difficult to follow due to its length and density. Break it into smaller, more focused paragraphs to improve readability. Ensure you thoroughly engage with the cited studies rather than discussing them superficially.
The limitations section is underdeveloped. Discuss additional limitations, their potential impact on results, and how they may have introduced bias.
Line 336: Replace "Multiple" with a more specific description.
Expand the discussion on how your results could inform treatment strategies or clinical practice. Provide practical insights into how the findings could be translated into improved patient care.
Ensure the reference list adheres to the journal's style guide.
Supplementary data should be presented in a separate file. However, demographic and clinical data comparing LN patients and controls should be included as a table in the main text for clarity and accessibility.
Comments on the Quality of English LanguageA native English speaker's review could enhance the paper's readability.
Author Response
Dear reviewer,
Thank you for your time and suggestions.
we made important modifications to the text (highlighted in yellow in the revised manuscript) and the figures.
We thank you again for giving us the opportunity to improve our work.
R3. Comment 1: The current title suggests the paper is more of a review than an original study. I recommend making the title more specific and informative to accurately reflect the content and focus of the study.
R3. Response 1: we agreed with this comment and changed the title to reflect our content.
“Decreased expression of aquaporin 3 as a feature of tubular damage in lupus nephritis”
R3. Comment 2: The "Results" section is challenging to follow. Consider presenting the results step-by-step to improve readability and clarity.
R3. Response 2: We clarify the flow of the results section as suggested.
R3. Comment 3: Adding information in the "Conclusions" section about the clinical utility of your findings, either as biomarkers or potential therapeutic targets, could enhance the abstract's impact and clinical relevance.
R3. Response 3: We thank the reviewer for this important note. The conclusion was modified as such: “We identified significant differences in the expression of aquaporins 1, 2, and 3 in patients with lupus nephritis. These findings strongly suggest that decreased AQP expression could serve as an indicator of tubular injury. Therefore, further research is warranted to evaluate AQP1, AQP2, and AQP3 as prognostic markers in both urinary and histological assessments of lupus nephritis.”
R3. Comment 4: In the first paragraph, emphasize the clinical differences between LN and non-LN SLE patients. Clearly articulate how these differences highlight the need for personalized treatment strategies in this subgroup.
R3. Response 4: We modified the introduction as suggested: “Systemic lupus erythematosus (SLE) is a heterogeneous disease, with significant variability in clinical presentation and organ involvement. Among its complications, lupus nephritis (LN) represents a distinct and severe subgroup, affecting up to one-third of SLE patients, with 5 to 20% progressing to end-stage kidney disease (ESKD). Compared to non-LN SLE patients, those with LN face a higher burden of morbidity, requiring more aggressive immunosuppressive therapy and closer long-term monitoring. Despite advances in our understanding of SLE, LN remains a major therapeutic challenge, with persistent gaps in predicting disease progression and response to treatment. These differences underscore the need for personalized treatment strategies tailored to the unique pathophysiology and prognostic factors of LN, optimizing both renal and overall patient outcomes.”
R3 Comment 5: Lines 44-46: Provide relevant references to support the data mentioned. Additionally, clarify whether "actors" or "factors" is the intended term.
Line 50: Include an appropriate reference for the statement made.
Line 60: Expand the first sentence to provide more context and include supporting references.
Use past tense consistently when describing the study's methodology (e.g., "we aimed to," "we analyzed").
Line 101: Provide the specific approval number and date for ethical approval.
Lines 105-106: Specify the source and conditions under which control kidney samples were collected.
Lines 109-118: Detail the specific methods used for measurements and analyses.
Line 157: State the type of test used to assess the normality of the data.
Line 167: Replace "Most" with "All except one," if accurate.
The "Statistics" section requires improvement. Clearly describe all statistical tests used, including their purpose and justification. For instance: Why were non-parametric tests used for correlation? Use Q1–Q3 ranges instead of "IQR."
Clarify whether the LN patients included were newly diagnosed SLE cases or those experiencing flares.
Table 1: Include the table in the main text rather than as a figure. Additionally, provide reference ranges for laboratory measurements if available.
Figure 2: Add specific p-values and ensure that statistically significant p-values are rounded to the third decimal place (or use p < 0.001 for very small values). Values > 0.05 should be rounded to the second decimal place.
Improve the readability of figures and tables. Consider using box-and-whisker plots (with min, max, median, Q1, and Q3) and overlaying dots for individual data points.
Line 279: Provide more references to substantiate claims about alterations in AQP.
Line 336: Replace "Multiple" with a more specific description.
Ensure the reference list adheres to the journal's style guide.
Supplementary data should be presented in a separate file. However, demographic and clinical data comparing LN patients and controls should be included as a table in the main text for clarity and accessibility.
R3 Response 5: We made the requested corrections.
R3 Comment 6: Confirm whether LN patients and controls were comparable in terms of sex and age.
R3 Response 6: We thank the reviewer for this interesting comment. We tested sex distribution by Fisher’s exact test and obtained a p-value of 0.33 confirming comparable sex distribution. For age distribution we performed a Mann-Whitney test and obtained significant older controls than lupus nephritis (p = 0.003). However, we tested correlation between age and AQP expression using Spearman correlation test and did not find any correlation, in addition literature in the field suggest a decreased expression with age in animal model which means if there is any effect of this differences it could underestimate the decrease observed in LN. We completed the limitations and perspective paragraph accordingly and add a graph of age distribution in supplementary files.
“Additionally, our study included a significant age difference between LN patients and controls. However, we found no correlation between AQP expression and age, and existing literature suggests that AQP expression might decrease with aging (26). If age had any effect, it would likely have led to an underestimation of the differences we observed”
R3 Comment 7: The discussion is difficult to follow due to its length and density. Break it into smaller, more focused paragraphs to improve readability. Ensure you thoroughly engage with the cited studies rather than discussing them superficially.
The limitations section is underdeveloped. Discuss additional limitations, their potential impact on results, and how they may have introduced bias.
Expand the discussion on how your results could inform treatment strategies or clinical practice. Provide practical insights into how the findings could be translated into improved patient care.
R3 Comment 7: We agreed with this comment and rewrite the discussion and limitation/perspectives paragraph.
Reviewer 4 Report
Comments and Suggestions for Authors
I appreciate the editors for the opportunity to peer-review the manuscript titled "Aquaporin 1, 2, and 3 expressions in Lupus Nephritis: Insights into tubular pathology." The manuscript is well-written, and the experiments are carefully conducted. Below are my comments for consideration:
- Would you be able to discuss any differences in the parameters analyzed with respect to sex? Does excluding 1 male patient impact any parameter studied?
- I suggest merging sections 3.2 and 3.3, as the discussion in Figure 2 is dependent on Figure 1. Combining these sections would improve the logical flow of the manuscript.
- Why was AQP1 expression in the cortex not analyzed according to control, NP-LN, and P-LN categories as it was in Figure 3A? Including this analysis would provide a more complete comparison.
- Line 286: Based on my understanding, the results in Reference 21 indicate an apparent loss of AQP1 staining on the basolateral membranes of the tubules in lupus nephritis biopsy specimens (Figure 3, h through j). However, the statement in that reference paper's summary section, claiming "uniformly increased expression of AQP-1, especially in glomeruli, in association with all forms of renal disease investigated," seems contradictory to the results discussed. Please verify this and revise the discussion section accordingly to ensure consistency and clarity.
Thank you for the opportunity to review this important work.
Author Response
Dear reviewer,
Thank you for your time and energy reviewing our work.
According to the recommendations of the referees, we made important modifications to the text (highlighted in yellow in the revised manuscript) and the figures.
please find below answers to your queries.
R4. Comment 1: Would you be able to discuss any differences in the parameters analyzed with respect to sex? Does excluding 1 male patient impact any parameter studied?
R4. Response 1: We thank the reviewer for this comment. We carefully reviewed each graph presented in the main article and highlighted males. We did not see any effect on our conclusion when removing males samples. We also clarify cases and controls clinical data and included it in the main text for easier comparison.
R4. Comment 2: I suggest merging sections 3.2 and 3.3, as the discussion in Figure 2 is dependent on Figure 1. Combining these sections would improve the logical flow of the manuscript.
R4. Response 2: We thank the reviewer for this suggestion and modified the manuscript accordingly.
R4. Comment 3: Why was AQP1 expression in the cortex not analyzed according to control, NP-LN, and P-LN categories as it was in Figure 3A? Including this analysis would provide a more complete comparison.
R4. Comment 3: The requested analysis was added to Figure 3A.
R4. Comment 4: Line 286: Based on my understanding, the results in Reference 21 indicate an apparent loss of AQP1 staining on the basolateral membranes of the tubules in lupus nephritis biopsy specimens (Figure 3, h through j). However, the statement in that reference paper's summary section, claiming "uniformly increased expression of AQP-1, especially in glomeruli, in association with all forms of renal disease investigated," seems contradictory to the results discussed. Please verify this and revise the discussion section accordingly to ensure consistency and clarity.
R4 Response 4: We agreed with the reviewer and detailed our statement about reference 21 in the discussion.
Round 2
Reviewer 1 Report
Comments and Suggestions for Authors
No.
Comments on the Quality of English LanguageGood
Author Response
There no comments to be adressed
Reviewer 3 Report
Comments and Suggestions for Authors
I would like to thank the authors for replying to my comments. Please find below my additional suggestions.
-
The title presented in the manuscript differs from the one presented in the response.
-
One of the limitations of the study is the relatively small sample of controls included in the analysis.
-
In the first paragraph of the Introduction section, please include the papers that you have cited (only two were mentioned at the end).
-
I suggest rewriting the Introduction in the past tense, for example: "We aimed to..."
-
There are some typos in the text. Please carefully review the entire manuscript.
Author Response
Dear reviewer ,
we thank you for the comments and suggestions . please find below our reponses to your queries :
The title presented in the manuscript differs from the one presented in the response.
1/ THE title has been modified according to the demand of another reviewer
One of the limitations of the study is the relatively small sample of controls included in the analysis
2/ We have implemented the fact that one of the limitations is the small sample size .
In the first paragraph of the Introduction section, please include the papers that you have cited (only two were mentioned at the end).
3/ We have included 2 more references in the text
I suggest rewriting the Introduction in the past tense, for example: "We aimed to..."
4/ the introduction has been rewritten in the past tense where appropriate
There are some typos in the text. Please carefully review the entire manuscript.
5/ We have reviewed the text and corrected for any typos errors .
Again , on behalf of the authors we thank you for your useful comments and suggestions
Reviewer 4 Report
Comments and Suggestions for Authors
The authors have addressed my comments in the manuscripts satisfactorily. The manuscript is acceptable in the current format.
Author Response
No comments to be addressed